# GUC: Unsupervised non-parametric Global Clustering and Anomaly Detection

## Abstract

Clustering is a crucial task in the fields of Machine and Representation Learning, with the ability of grouping similar data points being of particular importance. In this paper, we propose a novel non-parametric algorithm that performs Global Clustering and Anomaly Detection in an unsupervised manner. Our algorithm is both effective and efficient, requiring no prior assumptions or domain knowledge to be applied. It features two modes that utilize the distance from the dataset's center for clustering data points together. The first mode splits the dataset into global clusters where each cluster signifies proximity from the center. The second mode employs a threshold value for splitting data points into outliers and inliers. We evaluate our proposal against other prominent methods using synthetic and real datasets. Our experiments demonstrate that the proposed algorithm achieves state-of-the-art performance with minimum computational cost, and can successfully be applied to a wide range of Machine Learning applications.

## 1 Introduction

We begin our work by defining a vector $x = (x_1, x_2, \dots, x_d)$ to be a data point in the $d$-dimensional Euclidean space $x \in \mathbb{R}^d$ drawn *iid* from an unknown distribution $X$. Here, the pdf of the distribution is also unknown, hence the probability of assuming a particular point. In the context of outlier detection, a supervised clustering algorithm such as KNN would generally perform well for a set of data $X = \{(x_1, y_1), (x_2, y_2), \dots, (x_N, y_N)\}$ with known labels $y \in \{0, 1\}$, due to its ability to predict labels based on neighboring points. However, in unsupervised clustering the labels are now latent with $X$ being defined as $X = \{(x_1, \bullet), (x2, \bullet), \dots, (x_N, \bullet)\}$, and splitting the data globally means that each point belongs to a cluster that is yet to be observed. Therefore, each point needs to be assigned to a particular cluster with center $c \in C$ where $C = \{c_1, c_2, \dots, c_K\}$ and $K$ being a fixed quantity defined by the user. In our work we assume a non-binary classification of anomalies, and we aim to generate the set $C$ consisting of $K$ global centers, and assign each point to the nearest cluster. Global clusters and the distance from their centers are defined in the next sections. For each point $x \in X$ the probability mass function of belonging to a cluster from $\{c_1, c_2, \dots, c_K\}$ is defined by $\sum_{k=1}^{K} P(c_k|x) = 1$ where $P(c_k|x)$ needs to be estimated. From a probabilistic standpoint is difficult to estimate:

$$P(c_k|x) = \frac{P(x|c_k)P(c_k)}{P(c_1)P(x|c_1) + \dots + P(c_K)P(x|c_K)} = \frac{P(x|c_k)P(c_k)}{\sum_{k=1}^{K} P(c_k)P(x|c_k)}$$ since a distribu-

tional assumption would have to be made, and even so, the probability of observing a particular cluster is still unknown. We can however approximate $P(c_k|x)$ with a function that assigns the point to the closest cluster-center, i.e.: $\arg\max_k P(c_k|x) := f(x|C) = \arg\min_{c \in C}\{d(x, c)\}$, with $d(\cdot)$ representing a distance measure.

## 2 Background

We begin by reviewing parametric and non-parametric unsupervised clustering algorithms, and how they relate to our work. Gaussian Mixture Models (GMM -Dempster et al. (1977)), represent a probabilistic approach for modeling data distributions in unsupervised learning. Given a set of data points $\{x_i\}_{i=1}^{N}$, GMM assumes that the data points are generated from a mixture of $K$ Gaussian components. If the model parameters $\theta = (\pi_k, \{\mu_k, \sum_k\})$ for each cluster are known, the most probable cluster for a data point $x$ can be estimated by $c = \arg\max_k P(c_k|x, \theta) \propto \arg\max_k P(c_k|\theta)P(x|c_k, \theta)$. The parameters $\pi_k, \mu_k, \Sigma_k$ are estimated through methods like the Expectation-Maximization (EM) algorithm. Although K-means can also be derived as a special case of GMM, in which $\sum_k = I$ and $\pi_k = 1/K$, here we include the conventional distance-based definition of K-means clustering. K-means MacQueen et al. (1967) is an iterative cluster-

ing algorithm that partitions a dataset, and aims to assign a point $x$ to the closest cluster center, i.e.: $c = \arg\min_k \|x - \mu_k\|^2$, where $\mu_k$ represents the cluster center, given by the mean $\mu_k = \frac{1}{N_{c_k}} \sum_{i=1}^{N_{c_k}} x_i$ of within-cluster points. The process of assigning data points to clusters and updating the cluster centers is repeated until convergence. Other distance-based clustering algorithms such as PAM - Rdusseeun & Kaufman (1987), CLARANS - Ng & Han (2002), have been applied with some success to outlier detection (Murugavel & Punithavalli (2011); Vijayarani et al. (2011); Lei et al. (2012)) although none of these algorithms can be used for creating groups of anomalies and can only accommodate binary clustering (outliers or otherwise).

Unsupervised, density-based clustering algorithms are designed to identify dense regions in data while identifying points that are not part of any such density-based structure as outliers. DB-SCAN - Ester et al. (1996) aims to identify core points, border points, and noise points from a given set of data. It's governed by two hyperparameters epsilon ($\epsilon$) and $MinPts$, and is able to form a density if the conditions of Maximality ($\forall x, x'$ : if $x \in c$ and $x'$ is density reachable from $x \implies x' \in c$) and Connectivity ($\forall x, x' \in c$ : $x$ is density-connected to $x'$) hold true w.r.t. to $\epsilon$ and $MinPts$ values. LOF - Breunig et al. (2000) improves density-based clustering by quantifying the degree that each point is isolated from the surrounding densities, s.t. $LRD(x) = 1/(\frac{\sum_{x' \in N_k(x)} d_k(x,x')}{|N_k(x)|})$, where $d_k(\cdot)$ represents the Euclidean distance. LOF introduced the Outlier Factor score, $LOF(x) = \sum_{x' \in N_k(x)} \frac{LRD_k(x')}{LRD_k(x)} / |N_k(x)|$, whereas $LRD$ denotes the local reachability density, and $N_k(x)$ refers to the k-nearest neighbors of $x$. Similar approaches to LOF include COF - Tang et al. (2002), INFLO -Jin et al. (2006), LoOP - Kriegel et al. (2009), which use different distance measures or outlier scores for enhancing local density-clustering and anomaly detection.

The One-Class SVM Schölkopf et al. (1999) is a binary classification algorithm that generates a decision boundary around the majority of the data, identifying outliers as deviations from this boundary. Mathematically, oSVM aims to solve the optimization problem: $\arg\min_{w,b,\xi} \frac{1}{2}\|w\|^2 + \frac{1}{vN} \sum_{i=1}^{N} \xi_i - b$ subject to the constraints $w^T \phi(x_i) \geq b - \xi_i$ and $\xi_i \geq 0$, where $\phi(\cdot)$ represents the RBF kernel and $\xi_i$ a slack variable for dynamic margin formulation. Here $\nu$ (nu) represents the contamination parameter (proportion of outliers in the dataset) and controls the trade-off between maximizing the margin and minimizing the number of outlier-points.

Isolation Forest by Liu et al. (2008) comprises a fundamentally different approach for detecting anomalies. Similar to conventional random forests, iForest leverages binary tree structures to isolate anomalies by recursively partitioning the data space. It utilizes an anomaly score $S(x) = 2^{\frac{E(h(x))}{c(N)}}$ where $E(h(x))$ is the expected path length of $x$ in a tree, and $c(N)$ a normalization constant; for detecting anomalous data points that require fewer splits to be isolated. However, they have been shown to suffer from some bias due to the way the splits takes place (Hariri et al. (2019)).

Reviewing the existing solutions in the context of outlier detection uncovers their limitations. Some algorithms make distributional assumptions, and their performance is not guaranteed if the assumptions do not hold. Some are computationally intensive, unsuitable for big data. Distance-based algorithms do not allow for data points that are located far from each other to be clustered together, resulting in the anomalies being part of some of the formed clusters. Density-based algorithms, can group distant points together, but are designed primarily for identifying clusters of points in local densities. Although iForest does not suffer from any of the above it is often prone to biased results.

Our proposal addresses these limitations by being designed specifically for global clustering, not relying on distributional assumptions, allowing the grouping of distant points, consistently producing unbiased results, and having excellent computational efficiency.

## 2.1 DEFINTIONS

We begin our work by defining the necessary notions of Global Clustering and Global Outlier Detection. Existing distance-based algorithms assign points to local clusters based on the distance from their centroids. In the case of global clustering, the center needs to be in a single fixed location, for measuring the topological distance of each point in the data space.

**Definition 1 - Global Center:** The center of a dataset is given by the set of average values of each dimension: $\underline{\bar{x}} = \{\bar{x}_j\}_{j=1}^d$.

**Definition 2 - Global Clustering:** Given a set of global clusters centers $C = \{c_1, c_2, \dots, C_K\}$ a point can be assigned to a cluster based on the minimum distance from the clusters' center, i.e.: $c_x = arg\,min_{c \in C}\{d(x, c)\}$.

**Definition 3 - Global Outliers:** A point $x$ is classified as an inlier if $\|x - \underline{\bar{x}}\|_2 \leq q$ where $q$ is the threshold distance for a point to be considered an inlier.

## 3 GLOBAL UNSUPERVISED (GU)-CLUSTERING

In the context of Global Clustering, a point $x$ can be topologically located from the dataset's center with some finite distance. Here we are interested in generating synthetic labels $y \in \mathbb{R} \; \forall \; x \in X$ which indicate the proximity of each point from the center. Specifically using the Euclidean distance,

$$Y_i = \|x_i - \underline{\bar{x}}\|_2 = \sqrt{\sum_{j=1}^{d} \left(x_i^{(j)} - \bar{x}_j\right)^2} \tag{3.1}$$

we measure the global distance of each point from the center. This process generates the vector $Y$, a univariate distribution of point-distances. This distribution results from the fitting step of our proposal, as outlined in Algorithm 1. It represents the distribution of distances of each point from the center, which comprises the key component of our work.

---

**Algorithm 1:** Point-Distance from Global Center

**Data:** Input: $X \in \mathbb{R}^{N \times d}$
Initialize distance vector $Y = (0, 0, \dots, 0) \quad \forall x \in X$
**for** $j \leftarrow 1$ *to* $d$ **do**
  $\mid \quad Y = Y + (X_j - \bar{X}_j)^2$
**end**
$Y \leftarrow Y^{0.5}$

---

Furthermore, by utilizing the quantile discrepancy we can empirically measure the variability between a data point and a certain quantile. The vector of distances $Y$ from (3.1), results in a univariate random variable defined on the real line $\mathbb{R}$ with a strictly monotonically increasing CDF such that $q(\theta) = F_Y^{-1}(\theta) = inf\{y : F_Y(y) \geq \theta\}$ where $\theta \in [0, 1]$ is a percentile and $q(\theta)$ its corresponding quantile.

### 3.1 OUTLIER DETECTION

For performing outlier detection we can use the quantile discrepancy, and determine the value of $q(\theta)$ that minimizes the following variability measure:

$$\theta \int_{y > q(\theta)} |y - q(\theta)| \, \frac{d}{d_y} F_Y(y) + (1 - \theta) \int_{y < q(\theta)} |y - q(1 - \theta)| \, \frac{d}{d_y} F_Y(y) \tag{3.1.1}$$

Equation (3.1.1) can be evaluated empirically, and provides a practical measure for quantifying the spread of the distribution at certain quantiles, by:

$$var(Y) = \frac{\sum_{i=1}^{N} I(y_i > q(\theta))}{\sum_{i=1}^{N} I(y_i)} \sum_{i=1}^{N} \theta \, |y_i - q(\theta)| + \frac{\sum_{i=1}^{N} I(y_i \leq q(\theta))}{\sum_{i=1}^{N} I(y_i)} \sum_{i=1}^{N} (1 - \theta) \, |y_i - q(1 - \theta)| \tag{3.1.2}$$

where $\theta$ denotes the weight assigned to each quantile. Optimizing equation (3.1.2), we can determine the value of $\theta$ that leads to the best separation between inliers and outliers based on the corresponding quantile. However, it may not be feasible to solve equation (3.1.2) analytically, and numerical optimization methods such as projected gradient descent are necessary (since $\theta$ is constrained within the range [0,1]). While we have not pursued this optimization approach at this time, we leave this for future work.

The quantile discrepancy for a single point can be mapped as $f : \mathbb{R}^2 \rightarrow \{0, 1\}$, where $f(y, q(\theta)) = I(y > q(\theta))$ and for the vector of distances $Y$:

$$Y = f(Y, q(\theta)) = I(Y > q(\theta)) \tag{3.1.3}$$

where $q(\theta)$ denotes the threshold distance for a point to be considered an inlier. Thus, we generate binary labels $y \in \{0, 1\}$ to detect outliers based on their distance from the global center. Points that exceed a certain threshold distance are classified as outliers, while the remaining points are considered inliers.

---

**Algorithm 2:** Global Outlier Detection

---

**Data:** Vector of distances $Y = (y_1, y_2, \ldots, y_N)$, $q$
$Y \leftarrow I(Y > q)$

---

## 3.2 GLOBAL CLUSTERING

For global clustering the same approach is adjusted for measuring the variability between a data point and $K$ quantiles. Particularly, we aim to minimize the variability measure based on the number of chosen clusters $K$. Equation (3.1.1) can now be expressed as:

$$\sum_{k=1}^{K-1} (1 - \sum_{\substack{j=1 \\ j \neq k}}^{K} \theta_j) \int_{y \geq q(\theta_k)}^{q(\theta_{k+1})} |y - q(\theta_k)| \frac{d}{dy} F_Y(y) \, dy \tag{3.2.1}$$

Evaluating the integral, the overall variability of $Y$ can be measured by:

$$var(Y) = \sum_{k=1}^{K-1} (1 - \sum_{\substack{j=1 \\ j \neq k}}^{K} \theta_j) \, |q(\theta_{k+1}) - q(\theta_k)| \, P\left(q(\theta_k) \leq Y \leq q(\theta_{k+1})\right) \tag{3.2.2}$$

Equation (3.2.2) can be optimized to determine the optimal number of clusters that best split a dataset. Future work aims to explore techniques such as reducing in-cluster variability (similar to the elbow method in k-means), or using gradient descent for finding the optimal value of $K$. For measuring the overall variability of a single point:

$$\sum_{k=1}^{K-1} (1 - \sum_{\substack{j=1 \\ j \neq k}}^{K} \theta_j) \, |q(\theta_{k+1}) - y| \tag{3.2.3}$$

Following, we can estimate $K$ cluster centers by splitting $Y$ into $K$ quantiles, where the value of each quantile represents the center of each cluster.

$$C = \{c_1, c_2, \ldots, c_K\} := \{q(\theta_1), q(\theta_2), \ldots, q(\theta_K)\}$$

and for measuring the distance from each quantile (cluster-center):

$$\left\{ (1 - \sum_{\substack{j=1 \\ j \neq k}}^{K} \theta_j) \, |c_k - y| \right\} \forall \, c_k \in C \tag{3.2.4}$$

Assuming a choice of equally spaced percentiles, we can omit the weights of each quantile, and the cluster for each point in $Y$ can be determined based on the shortest distance from each cluster-center, such that:

$$c_x = \underset{k \in \{1, \ldots, K\}}{\arg \min} \{|c_k - y|\} \tag{3.2.5}$$

Thus, we can assign the points to the closest cluster by measuring the distance from each cluster-center. Measuring the absolute distance from the center of each cluster, we are now able to group points from both ends of the data space, and permit them to be in the same cluster. A practical implementation of Global Clustering is presented in Algorithm 3.

GU-Clustering performs component-wise operations which are computationally efficient and allow for vectorized operations. These operations result in very fast performance with a runtime complexity of O(d) for fitting the model, O(K) in the Global Clustering mode, and O(1) in the Outlier Detection mode. A comprehensive review of computational performance is featured in **Appendix A**.

---

**Algorithm 3:** Assigning Points to $K$ Global Clusters

---

**Data:** Vector of distances $Y = (y_1, y_2, ..., y_N)$, $K$

Calculate $K$ quantiles from $Y$

$Y \leftarrow \mathrm{argmin}_k \; abs((q_1, q_2, ..., q_K) - Y)$

---

# 4 EXPERIMENTS AND RESULTS

## 4.1 GLOBAL CLUSTERING IN SYNTHETIC DATASETS

The proposed notion of Global Clustering is applied to low-dimensional synthetic datasets for visual inspection. Common metrics used to evaluate the quality of an unsupervised clustering algorithms, such as the Silhouette Coefficient -Rousseeuw (1987) and Dunn's index - Dunn (1974) are not applicable in global clustering since they measure the distance from within-cluster points. Therefore in the context of global clustering, we measure the average distance of the points of each cluster from the dataset's center,

$$\left\{ \frac{1}{N_{c_1}} \sum_{i=1}^{N_{c_1}} Y_i, \; \frac{1}{N_{c_2}} \sum_{i=1}^{N_{c_2}} Y_i, \; ......, \; \frac{1}{N_{c_K}} \sum_{i=1}^{N_{c_K}} Y_i \right\} \tag{4.1}$$

where $Y_i$ as defined by equation $(3.1)$. This allows us to perform Global Clustering and quantify the rank of each point as an outlier.

### 4.1.1 SYNTHETIC DATASET 1. A 2-DIMENSIONAL SYNTHETIC DATASET WITH A SINGLE DENSITY

| Cluster | K-Means | DSCAN | GMM | GU-Clustering |
|---------|---------|--------|--------|---------------|
| 1 | 0.3289 | 0.3653 | 0.4896 | 0.0695 |
| 2 | 0.3167 | 0.1610 | 0.374 | 0.1911 |
| 3 | 0.3429 | 0.2614 | 0.2742 | 0.3261 |
| 4 | 0.3977 | 0.1733 | 0.2806 | 0.5341 |

Table 1: Average distance from the global center

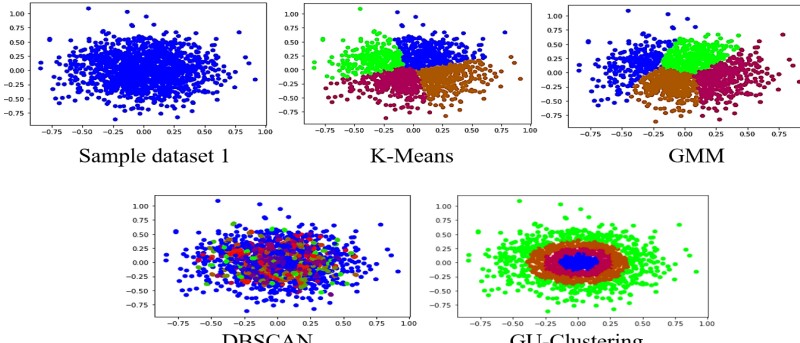

Sample dataset 1     K-Means     GMM

DBSCAN     GU-Clustering

Figure 1: For the algorithms that allow cluster selection $K$ was set to 4. DBSCAN generated a large number of clusters and does not produce meaningful results for either local or global clustering. GMM and K-Means split the data into 4 local clusters. GU-Clustering successfully splits the points into 4 global clusters.

### 4.1.2 SYNTHETIC DATASET 2. A 2-DIMENSIONAL SYNTHETIC DATASET COMPRISED OF 4 DISTINCT DENSITIES WITH THE SAME CENTER

| Cluster | K-Means | DSCAN | GMM | GU-Clustering |
|---------|---------|-------|-----|---------------|
| 1 | 690.53 | 70.590 | 784.78 | 70.590 |
| 2 | 693.70 | 301.29 | 777.62 | 301.29 |
| 3 | 639.82 | 548.99 | 410.71 | 548.99 |
| 4 | 486.48 | 900.73 | 696.20 | 900.73 |

Table 2: Average distance from the global center

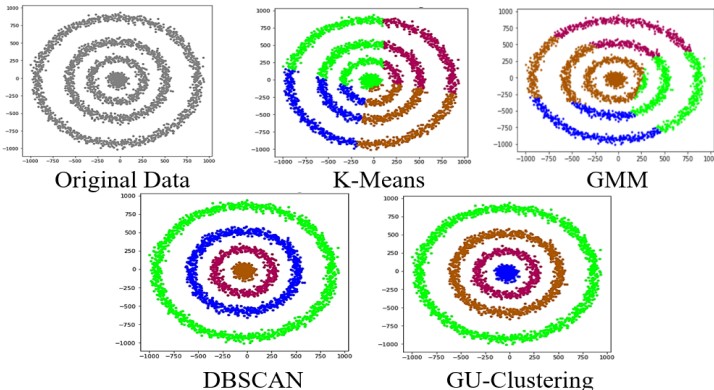

Figure 2: In this implementation, the number of clusters $K$ was set to 4. DBSCAN and GU-Clustering were successful in allocating the data to their correct densities, achieving the same distances from the center. GMM and K-means group the points locally and do not allow for distant points to share the same cluster.

### 4.1.3 SYNTHETIC DATASET 3. A 2-DIMENSIONAL SYNTHETIC DATASET WITH 4 DENSITIES AND 4 DISTINCT CENTERS

| Cluster | K-Means | DSCAN | GMM | GU-Clustering |
|---------|---------|-------|-----|---------------|
| 1 | 4.003 | 4.881 | 3.993 | 0.782 |
| 2 | 8.562 | 8.438 | 8.562 | 2.376 |
| 3 | 5.667 | 1.421 | 5.667 | 4.788 |
| 4 | 1.284 | 1.837 | 1.280 | 8.001 |

Table 3: Average distance from the global center

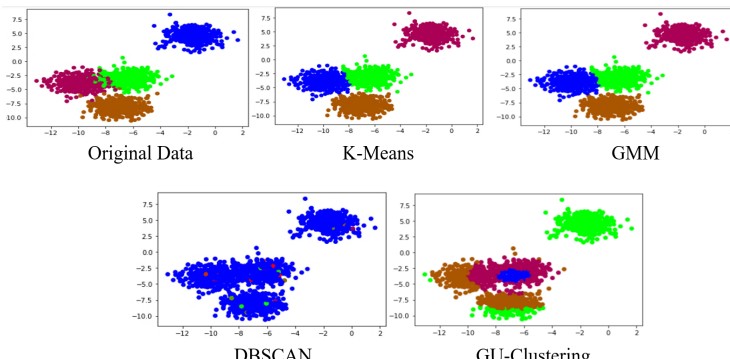

Figure 3: Similar to the previous applications, the number of clusters $K$ was set to 4, for the algorithms that allow cluster selection. GMM and K-means split the points into 4 local clusters, while GU-Clustering splits the points into 4 global clusters. DBSCAN was again difficult to calibrate, generating several arbitrary clusters.

### 4.1.4 SYNTHETIC DATASET 4. GLOBAL CLUSTERING IN A SYNTHETIC DATASET OF TIME SERIES DATA

| Cluster | K-Means | DSCAN | GMM | GU-Clustering |
|---------|---------|-------|------|---------------|
| 1 | 4,715 | N/A | 4,087 | 2,532 |
| 2 | 5,796 | N/A | 7,028 | 7,062 |
| 3 | 6,819 | N/A | 4,677 | 12,988 |
| 4 | 4,260 | N/A | 5,789 | 26,450 |

Table 4: Average distance from the global center

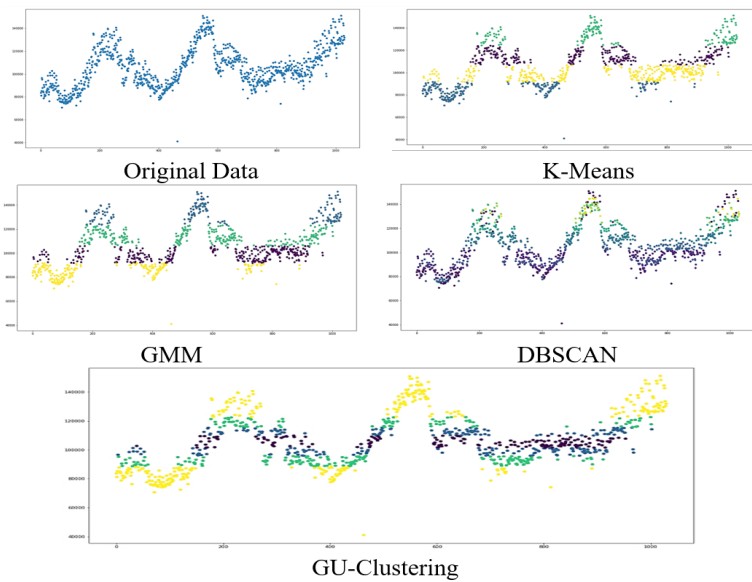

Figure 4: This paradigm illustrates the effectiveness of GU-Clustering for grouping points located on opposite ends of the data distribution. On the other hand, the competing algorithms, when confronted with the same time series data, still produce local clusters and fail to group anomalies from both ends of the distribution.

### 4.1.5 SYNTHETIC DATASET 5. A 3-DIMENSIONAL SYNTHETIC DATASET WITH 3 MIXED DENSITIES

| Cluster | K-Means | DSCAN | GMM | GU-Clustering |
|---------|---------|-------|-------|---------------|
| 1 | 1.580 | 2.032 | 1.459 | 0.543 |
| 2 | 1.648 | 1.001 | 1.641 | 1.170 |
| 3 | 1.611 | 1.521 | 1.725 | 2.166 |

Table 5: Average distance from the global center

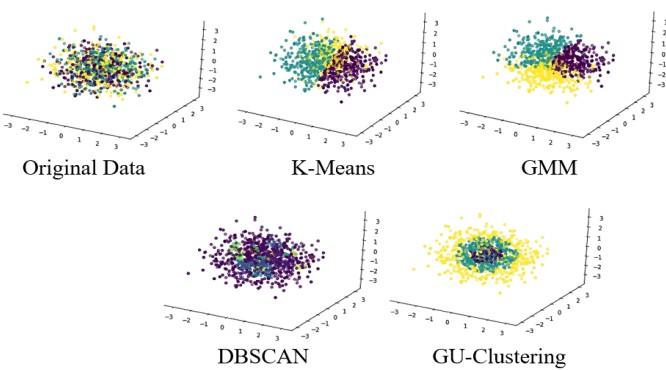

Figure 5: Similar to the previous paradigms, the conventional unsupervised clustering algorithms split the data locally, with DBSCAN generating a large number of non-systematic clusters.

The conventional unsupervised clustering algorithms fail to split the data globally, as they are optimized for local clustering and cannot group distant points together. DBSCAN is effective for handling arbitrary shapes but proved difficult to calibrate. It often exhibits an unsystematic behavior, generating a much larger number of clusters, since it does not permit the user to set the value of K. On the other hand, GU-Clustering excels in splitting the datasets globally, making it ideal for analyzing high-dimensional datasets where data visualization is impractical.

## 4.2 OUTLIER DETECTION

Although unsupervised anomaly detection does not utilize any label information, in this context they are needed for evaluation and comparison. The following datasets as presented by Goldstein et.al. Goldstein & Uchida (2016), are used for comparative evaluation of unsupervised anomaly detection algorithms. These datasets come from various domains and they differ in size, outlier percentage, and dimensionality, offering a broad evaluation spectrum. For outlier detection, the rank of the anomalies should be considered. Specifically, each point should be ranked as being an outlier compared against the other data points that comprise the dataset. In principal, a random outlier-point should rank higher than a random inlier-point. This is important in quantifying the quality (discrimination) of an algorithm. The Area Under the Curve (AUC) of the Receiver Operating Characteristic (ROC) curve is used to measure discrimination quality, with a focus on ranking using the AUC score. For evaluating predictive capabilities, the $F_1$ score is employed due to the inherent class imbalance of outlier prediction datasets.

| Dataset | Size | Dimensions | Outliers | Percentage |
|---|---|---|---|---|
| b-cancer - Lichman et al. (2013) | 367 | 30 | 10 | 2.72% |
| pen-global - Lichman et al. (2013) | 809 | 16 | 90 | 11.12% |
| letter - Lichman et al. (2013) | 1,600 | 32 | 100 | 6.25% |
| speech - Brümmer et al. (2012) | 3,686 | 400 | 61 | 1.65% |
| sattelite - Lichman et al. (2013) | 5,100 | 36 | 75 | 1.47% |
| per-local - Lichman et al. (2013) | 6,724 | 16 | 10 | 0.15% |
| ann-thyroid - Lichman et al. (2013) | 6,916 | 21 | 250 | 3.61% |
| shuttle - Lichman et al. (2013) | 46,464 | 9 | 878 | 1.89% |
| aloi - Geusebroek et al. (2005) | 50,000 | 27 | 1,508 | 3.02% |
| kdd99 - Lichman et al. (2013) | 620,098 | 29 | 1,052 | 0.17% |

Table 6: Datasets for Benchmarking

| Dataset | Robust Covariance | oSVM | iForest | LOF | GU-Clustering |
|---|---|---|---|---|---|
| b-cancer | 0.794 | 0.603 | 0.795 | 0.846 | **0.988** |
| pen-global | 0.601 | 0.660 | 0.750 | 0.706 | **0.945** |
| letter | 0.557 | 0.544 | 0.509 | **0.701** | 0.469 |
| speech | 0.5 | 0.505 | 0.508 | **0.517** | 0.492 |
| sattelite | 0.756 | 0.679 | 0.783 | 0.601 | **0.993** |
| pen-local | 0.5 | 0.5 | 0.5 | 0.5 | 0.5 |
| annthyroid | **0.672** | 0.502 | 0.527 | 0.492 | 0.482 |
| shuttle | 0.846 | 0.731 | **0.978** | 0.516 | 0.927 |
| aloi | 0.498 | 0.517 | 0.5 | 0.563 | **0.605** |
| kdd99 | 0.502 | 0.665 | 0.761 | 0.501 | **0.839** |

Table 7: Comparison of the discriminative performance of each outlier detection algorithm. GU-Clustering performs well, yielding the highest AUC score (higher is better) in 5 of the 10 datasets that were used in this comparison. The pen-local dataset proved to be difficult with all algorithms achieving exactly the same AUC score.

The most prominent algorithms for outlier detection are compared against our proposal for measuring their performance. In our experiments, the algorithms were implemented using the scikit-learn library Pedregosa et al. (2011). All of their hyperparameters were left at default with the contamination parameter set as the outlier percentage from Table 6 for achieving the best possible performance. For GU-Clustering, the hyperparameter $q$ was set to 0.999. The value at $q$ signifies the threshold distance for points to be classified as inliers. Preprocessing included applying a min-max transformation so the scale of all dimensions was between [0,1]. No other operations were performed and the datasets were used the same way as presented by Goldstein & Uchida (2016).

| Dataset | Robust Covariance | oSVM | iForest | LOF | GU-Clustering |
|---|---|---|---|---|---|
| b-cancer | 0.978 | 0.972 | 0.989 | **0.992** | 0.988 |
| pen-global | 0.842 | 0.924 | **0.944** | 0.935 | 0.942 |
| letter | 0.896 | 0.942 | 0.939 | 0.963 | **0.967** |
| speech | 0.967 | 0.976 | 0.984 | 0.984 | **0.991** |
| sattelite | 0.986 | 0.991 | **0.994** | 0.988 | 0.993 |
| pen-local | 0.997 | 0.997 | **0.999** | **0.999** | **0.999** |
| annthyroid | 0.954 | 0.964 | 0.966 | 0.963 | **0.981** |
| shuttle | 0.989 | 0.990 | **0.999** | 0.982 | 0.991 |
| aloi | 0.941 | 0.971 | 0.970 | 0.974 | **0.985** |
| kdd99 | 0.997 | **0.999** | **0.999** | 0.998 | **0.999** |

Table 8: Comparison of the predictive performance of each outlier detection algorithm. In terms of $F_1$ score (higher is better) GU-clustering performs the best by either leading the results or being a close second.

| Dataset | Robust Cov. | oSVM | iForest | LOF | GU-Clustering |
|---|---|---|---|---|---|
| b-cancer | 0.13s | 0.00s | 0.31s | 0.02s | 0.00s |
| pen-global | 1.12s | 0.02s | 0.34s | 0.02s | 0.00s |
| letter | 2.37s | 0.06s | 0.39s | 0.12s | 0.00s |
| speech | 65.9s | 0.46s | 2.58s | 0.54s | 0.01s |
| sattelite | 5.58s | 0.13s | 0.77s | 0.63s | 0.00s |
| pen-local | 4.67s | 0.09s | 0.81s | 1.24s | 0.00s |
| annthyroid | 1.18s | 0.41s | 0.80s | 1.16s | 0.00s |
| shuttle | 21.9s | 5.90s | 3.61s | 5.09s | 0.01s |
| aloi | 11.5s | 22.4s | 5.25s | 36.7s | 0.01s |
| kdd99 | 161s | 0.79hrs | 50.9s | 2.13hrs | 0.14s |

Table 9: Computational performance comparison of the algorithms evaluated in section 4.2. The experiments were conducted on a Google Colab environment with an Intel Xeon(R) CPU (1 core @ 2.2 GHz) and 12 GB of memory. GU-Clustering is more frugal on resources and significantly more efficient in terms of computational time.

## 5 CONCLUSIONS

Global Clustering is a critical process in learning representations and plays a vital role in applications such as data mining, anomaly detection, and pattern recognition. Despite its importance, past work in this area has not fully explored this potential. In turn, we address the limitations of the current solutions by providing a novel and effective framework for both global clustering and anomaly detection. Using quantiles to split the distribution into K regions, each point is assigned to the value of the closest quantile (defined as the cluster center in the global clustering mode) or classified as an outlier if it exceeds the cutoff distance (outlier detection mode). Our proposal is computationally efficient, has no underlying assumptions, and can be applied to a wide range of datasets. Finally, we evaluated the performance of our algorithm in various scenarios, and the results demonstrate its superiority over existing methods.

Regarding implementation, GU-Clustering is straightforward to implement, due to the few hyperparameters that govern it. However, similar to other algorithms that use distance to group data points, GU-Clustering might be affected by the scale of particular variables and a preprocessing step could be necessary (e.g. normalization, standardization) for producing unbiased results.

The work performed in this paper sets the foundation for future work that we want to carry out. Specifically, we aim to establish some well-grounded heuristics for choosing the optimal hyperparameters of our algorithm.

***The implementation of our algorithm in code with all the experiments and results can be reproduced in this anonymous GitHub repository.***
***https://github.com/SampleUser122/GU-Clustering***

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

APPENDIX A - EVALUATION OF COMPUTATIONAL PERFORMANCE

We now review the computational performance of the algorithms that are most prevalent in unsupervised clustering and are featured in this work, both for global clustering, and outlier detection. GU-Clustering operates in two modes, a Global Clustering mode, and a Global Anomaly Detection mode. It is designed to be simple to implement, yet highly effective and efficient. Since the data are stored in contiguous blocks of memory, it allows to perform component-wise operations. Component-wise operations are computationally efficient, and allow for vectorized operations, rather than looping over each element of the data structure individually.

| Algorithm | Run time complexity |
|---|---|
| K-means | $O(nKdi)$ |
| GMM | $O(nKd^3)$ |
| FMCD (RC) | $O(nlog(n))$ |
| DBSCAN | $O(nlog(n))$ |
| LOF | $O(n^2) + O(nlog(n))$ |
| oSVM | $O(n^2)$ |
| iForest | $O(n)$ |
| GU-Clustering | $O(d)$ |

Table 10: Run Time Complexity of featured unsupervised clustering algorithms

Although there are various implementations of the algorithms that are presented above, we review the variants used for benchmarking our proposal. Generally, algorithms that use covariance matrices to uncover the relationships between points tend to be slow, since a run time complexity of $O(n^3)$ is required for computing the covariance matrix, and cannot be applied efficiently to large datasets. Despite that, it is worth mentioning that in the case of oSVM, the run time complexity reduces to $O(n^2)$ when the contamination parameter is provided. FMCD (Robust Covariance) is more efficient since it uses a sample of the dataset for estimating the covariance matrix. Traditional K-means needs to sum the distance for $d$ dimensions, for $n$ points, and check for $K$ clusters if the point can be included. Usually, it takes a number of $i$ iterations to converge to the optimal result. Depending on the implementation of the algorithm this can be reduced to a time complexity of $O(nKi)$. iForest is the most efficient among the competing algorithms operating in linear time for the evaluation of each point. DBSCAN and LOF also operate in a polynomial run time complexity, for visiting every neighboring region and testing each point as a candidate for that region. LOF however, does default to a higher run time complexity, since it needs to compute the score for each data point and work out the upper/lower bounds for its minpts parameter. GU-Clustering is by far the most efficient algorithm since all its operations are performed component-wise, and requires only $d$ iterations for fitting the model. In the evaluation process, the run time complexity is reduced to $O(K)$ for the global clustering mode and $O(1)$ for the outlier detection mode.

| Dataset | Robust Cov. | oSVM | iForest | LOF | GU-Clustering |
|---|---|---|---|---|---|
| b-cancer | 0.13s | 0.00s | 0.31s | 0.02s | 0.00s |
| pen-global | 1.12s | 0.02s | 0.34s | 0.02s | 0.00s |
| letter | 2.37s | 0.06s | 0.39s | 0.12s | 0.00s |
| speech | 65.9s | 0.46s | 2.58s | 0.54s | 0.01s |
| sattelite | 5.58s | 0.13s | 0.77s | 0.63s | 0.00s |
| pen-local | 4.67s | 0.09s | 0.81s | 1.24s | 0.00s |
| annthyroid | 1.18s | 0.41s | 0.80s | 1.16s | 0.00s |
| shuttle | 21.9s | 5.90s | 3.61s | 5.09s | 0.01s |
| aloi | 11.5s | 22.4s | 5.25s | 36.7s | 0.01s |
| kdd99 | 161s | 0.79hrs | 50.9s | 2.13hrs | 0.14s |

Table 11: Computational performance comparison of the algorithms evaluated in section 4.2. The experiments were conducted on a Google Colab environment with an Intel Xeon(R) CPU (1 core @ 2.2 GHz) and 12 GB of memory. GU-Clustering is more frugal on resources and significantly more efficient in terms of computational time.

## APPENDIX B - VISUAL INSPECTION WITH LEADING ANOMALY DETECTION ALGORITHMS

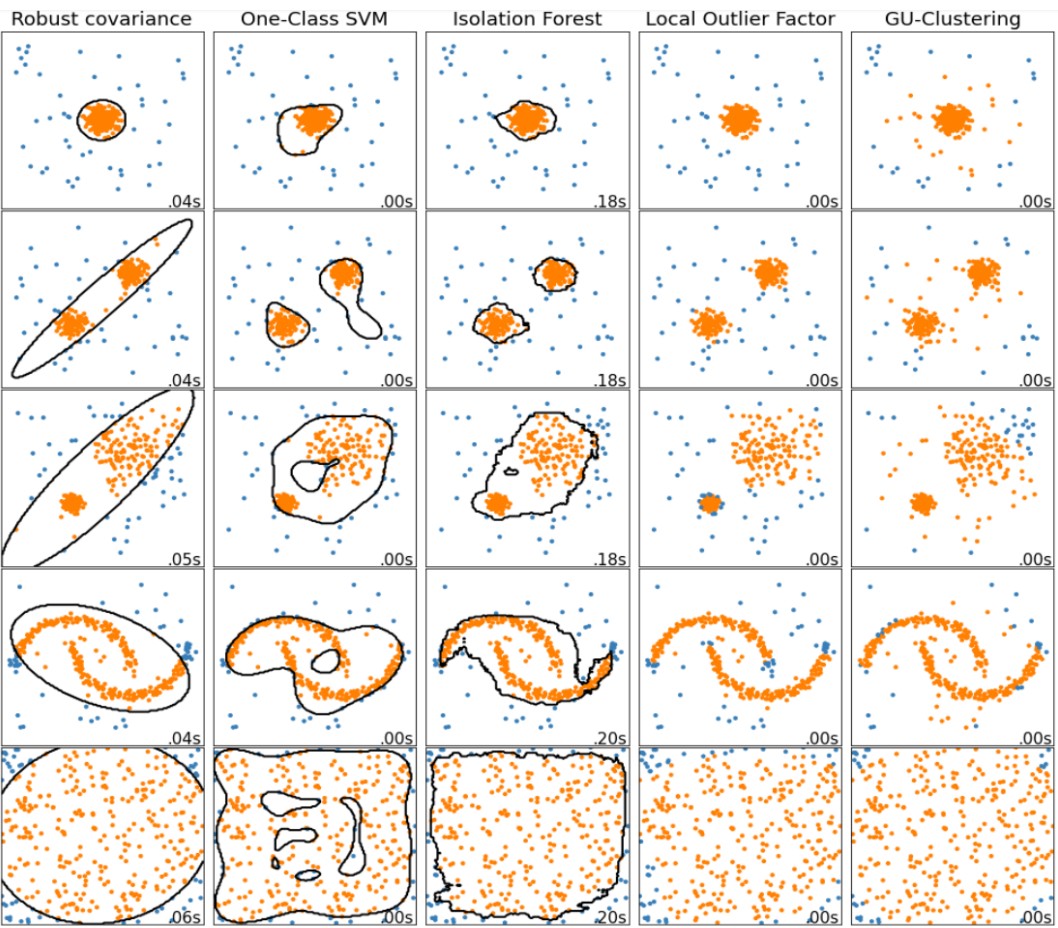

Figure 6: This application is an extension of scikit-learn's section of comparing outliers for anomaly detection Pedregosa et al. (2011). The datasets contain one or two modes (regions of high density) to illustrate the ability of the algorithms to cope with multimodal data. For each dataset, 15% of samples are generated as random uniform noise. This proportion is the value given to the nu parameter of the oSVM and the contamination parameter of the other outlier detection algorithms. The decision boundaries between inliers and outliers are displayed in black except for LOF and GU-Clustering as they do not have a predict method to be applied to new data. All algorithms parameters were hand-picked by the authors for achieving the best results. GU-Clustering effectively detects global outliers, as seen in the figure above, providing a new perspective to outlier detection. Similar to the results from the previous sections, the outliers detected focus on the global position of the points rather than outliers belonging to specific densities. Similar to LOF, GU-Clustering does not need to be fitted to a dataset before classifying new points, thus offering the possibility of detecting outliers in real time. Furthermore, GU-Clustering does not require prior knowledge of the contamination fraction and can achieve state-of-the-art performance using a default value for $q$.

