# OpenReview forum: "GUC: UNSUPERVISED NON-PARAMETRIC GLOBAL CLUSTERING AND ANOMALY DETECTION"
_ICLR.cc/2024/Conference — ICLR 2024 Conference Withdrawn Submission_

### Official Review · Reviewer_XPjK · 2023-10-28

**Soundness:** 2 fair
**Presentation:** 2 fair
**Contribution:** 2 fair
**Rating:** 3
**Confidence:** 5

**Summary:**

The paper proposes GUC for global clustering and global outlier detection. In short, GUC detects clusters and outliers based on distances of data samples to a global center, defined by the set of average dimension values.

**Strengths:**

The method is simple and straightforward to implement.

**Weaknesses:**

The global center and $Y$ distance vector are both susceptible to outliers. Hence, threshold $q(\theta)$ for outlier detection and thresholds $q(\theta_k)$ are not robust to extreme values.

It is not clear how to identify for $q$ in outlier detection and $q_k$ in clustering.

It is also not clear how to derive cluster center $c_k$ from $q_k$. Such center is important for result interpretability.

In addition, GUC is prone to the curse of dimensionality since it uses distances computed in the full-dimensional space.

On synthetic datasets not having center point exactly at the center, GU-clustering didn't perform well.

For outlier detection on real-world data sets, GU-clustering also didn't truly outperform baselines.

**Questions:**

A theoretical analysis of robustness of $q$ and $q_k$ to outliers will make the claims more convincing.

---

### Official Review · Reviewer_mhaU · 2023-11-01

**Soundness:** 2 fair
**Presentation:** 1 poor
**Contribution:** 2 fair
**Rating:** 3
**Confidence:** 4

**Summary:**

The paper proposes a non-parameter solution to perform clustering and anomaly detection in an unsupervised manner. The first idea focuses on splitting the dataset into global clusters, whereas the second idea requires a threshold to separate outliers from the normal data behavior. Experimental results demonstrate the performance of the proposed solution for two tasks, clustering and anomaly detection.

**Strengths:**

- Important problem, this is a well-studied area for both clustering and anomaly detection solutions

**Weaknesses:**

- Writing needs significant improvement
- Novelty is unclear
- Tons of baselines missing from clustering and anomaly detection areas

**Questions:**

- Writing needs significant improvement

The paper needs heavy re-writing. It reads as a "math" paper and does not follow how traditional papers appear at ML/AI venues. There is need for motivation/preliminaries/differences to prior solutions/related works etc.

- Novelty is unclear

The paper jumps into the solution without introducing any preliminaries for prior works. Is as if this is the first clustering paper. The baselines/concepts used are text-book material, again, as if someone has ignored thousands of papers for clustering and anomaly detection. It's unclear what is trully novel.

- Tons of baselines missing from clustering and anomaly detection areas

Both areas are enormous, with thousands of papers, and potential baselines. The paper compares with methods found in textbooks, which is "intro to ML/data mining" level material. Datasets in most cases are synthetic when we have again dozen of real-world datasets.

The paper needs improvements in every way, writing, stronger experiments, experimental settings, clarifying novelty, stronger/recent baselines, more real-world datasets.

---

### Official Review · Reviewer_mrV6 · 2023-11-05

**Soundness:** 1 poor
**Presentation:** 1 poor
**Contribution:** 1 poor
**Rating:** 1
**Confidence:** 5

**Summary:**

This paper presents a clustering and anomaly detection strategy in which cluster centers are fixed, and distance based thresholding is used to determine a (spherical) boundary between inliers and outliers, or between different shells as clusters.

**Strengths:**

S1) The minimization of a standard weighted quantile discrepancy measure is proposed as a way of setting a cluster radius (for outlier determination) or shells of low variability with a common center (for clustering into shells). These radii set a threshold by which inliers and outliers are separated, or different shells are separated as clusters.

**Weaknesses:**

W1) The idea of determining outlierness using distance-based thresholding from cluster centers is not novel. Further, determining clusters as rings or shells from the center of mass also falls well below the standard of novelty required of research contributions at this level.

W2) The authors fail to make a case as to why their assumptions regarding the shape and nature of clusters may be warranted.

W3) The topic and techniques proposed do not involve learning representation per se, and as such would not be well-suited for ICLR.

W4) The authors appear not to be following through on their proposed use of quantile discrepancy (citing it as future work), and seem to be choosing the quantiles arbitrarily.

W5) The presentation and organization is quite poor. The objectives are not clearly stated, and it takes several pages before they can be inferred.

**Questions:**

Please address W1 through W5.